# Sclerotherapy on Demand with Polidocanol to Treat HHT Nosebleeds

**DOI:** 10.3390/jcm10173845

**Published:** 2021-08-27

**Authors:** Sol Marcos, Luisa María Botella, Virginia Albiñana, Agustina Arbia, Anna María de Rosales

**Affiliations:** 1Otorrhinolaringology Department, Hospital Universitario Fundación Alcorcón, 28922 Madrid, Spain; roagusdi@gmail.com; 2CIBER Rare Diseases Unit 707, Centro de Investigaciones Biológicas Margarita Salas, CSIC, 28040 Madrid, Spain; cibluisa@cib.csic.es (L.M.B.); vir_albi_di@yahoo.es (V.A.); 3Pharmaceutical Department, Hospital Universitario Fundación Alcorcón, 28922 Madrid, Spain; annamariamrc@gmail.com

**Keywords:** HHT, epistaxis, sclerotherapy, polidocanol, propranolol, HHT-ESS, quality of life

## Abstract

Epistaxis is the most prevalent clinical symptom in Hereditary Haemorrhagic Telangiectasia (HHT), causing anaemia and decreasing the quality of life (QOL). Since 2013, in Hospital Universitario Fundación Alcorcón, more than 150 HHT patients have been treated by nose sclerotherapy on demand. This study shows the results of 105 patients treated with sclerotherapy between 2017 and 2019. HHT-ESS (epistaxis severity score) was used to measure the severity and frequency of epistaxis. QOL was determined before and after treatment by EuroQol-5D (EQ-5D) and the visual analogue scale (VAS) on the health condition. According to HHT-ESS before treatment, 22 patients presented mild, 35 moderate, and 47 severe epistaxes. Sclerotherapy significantly decreased the frequency and severity of epistaxis, with a significant drop of HHT-ESS in 4.6 points, from 6.23 ± 2.3 to 1.64 ± 1.6. Furthermore, the QOL significantly improved, the EQ-5D scale raised from 0.7 ± 0.26 pre- to 0.92 ± 0.16 post-treatment (*p* < 0.05). Additionally, VAS mean value showed a significant increase from 4.38 ± 2.4 to 8.35 ± 1.2. The QOL improvement was correlated with the ESS decrease. In conclusion, this study shows that on-demand sclerotherapy at the office significantly reduces HHT epistaxis as well as improved the patients’ QOL.

## 1. Introduction

Hereditary haemorrhagic telangiectasia (HHT), also known as Rendu–Osler–Weber disease, is an autosomal dominant multisystemic vascular disorder with incomplete penetrance. The estimated prevalence ranges from 1:5000 to 1:8000 [1], and it is thus considered a rare disease.

HHT affects several organs and is therefore the cause of a wide variety of clinical manifestations. The diagnosis for HHT is achieved by the Curaçao criteria [2]: epistaxis, mucocutaneous telangiectases, first-degree family inheritance, and visceral arteriovenous malformations (AVM). The presence of three of these four criteria results in a conclusive HHT diagnosis. Positive genetic tests are definitive for the diagnosis, being especially useful for young patients that have not yet developed clinical manifestations, according to Curaçao criteria [3]. Affected individuals display a variety of vascular malformations from telangiectases (dilated and very fragile capillaries) to AVMs. Telangiectases are usually present in the nasal mucosa but also on the lips, tongue, and on the tip of the fingers. There are vascular lesions in the gastrointestinal (GI) tract, AVMs on the liver, lungs, and more rarely (less than 10%) on the Central Nervous System. Typical lesions exhibit markedly dilated, tortuous venules with absent or abnormal muscle walls, frequently directly linked to unusually dilated arterioles [4]. On the nasal epithelium, telangiectases with thin walls are exposed to breathing air trauma, leading to dryness and nasal crusting. Damage causes rupture of these vessels and the lack of functional elastic fibres prevents normal vasoconstriction; thus, haemorrhage becomes difficult to handle. In an online survey of 666 HHT patients, 97% stated daily nosebleeds, and 49% received invasive treatments to control the haemorrhages [4]. Continuous bleeding can result in anaemia with the need for iron therapy and eventually blood transfusions.

In another questionnaire, among 220 HHT patients, about 50% suffered from daily bleeding, and 76.5% had nosebleeds at least once a week [5]. Nose bleeding in HHT requires quick actions, from nasal packing with absorbable materials to more invasive procedures and even hospitalisations [5]. Nosebleeds severity often becomes worse with age, impairing a patient’s normal life. Chronic anaemia leads to continuous visits to the hospital, blood transfusions, invasive procedures, and hospitalisations. All these situations have a significant detrimental impact on the HHT patients’ quality of life (QOL) [6].

Epistaxis is the most frequent clinical manifestation (more than 90% of HHT patients). Prevention for mild recurrent bleeding requires careful cleaning and moisturising of nasal nostrils. A variety of topical and oral drugs have been used for therapy: hormones such as estriol; selective estrogen receptor modulators such as tamoxifen or raloxifene; antifibrinolytics such as tranexamic acid (showing anti-inflammatory proprieties); and antiangiogenic agents such as propranolol or bevacizumab [7]. HHT management needs multidisciplinary assistance. Recurrent or severe epistaxis should be referred to an otorhinolaryngologist for local control and followed by blood tests to evaluate the need for oral or IV iron supplementation or blood transfusions to treat anaemia.

Management of epistaxis in HHT patients is different from non-HHT. A nosebleed can be the most predominant and incapacitating symptom in the elderly. Unfortunately, there is no agreement on which is the best type of treatment. Nasal packing can lead to more severe bleeding when removed, and cauterisation not only can worsen the lesions but also carries a risk of septal perforation. There are numerous treatments, including laser, radiofrequency, sclerotherapy, embolisation, and surgical procedures. Nevertheless, none is perfect, and most of them show limited efficacy, especially in severe cases.

Different treatments are applied depending on the frequency and severity of epistaxis. A nosebleed is often an emergency, but its management is also determined by the clinician’s experience. For recurrent nose bleedings, ENT specialists can perform several forms of laser photocoagulation (CO_2_, argon, neodymium-doped yttrium aluminium garnet (Nd-YAG), flashlamp-pulsed dye, and potassium titanyl phosphate (KTP)), plasma coagulation or surgical micro-debridement. More invasive surgery options involve septodermoplasty and full nasal closure (Young’s procedure) [3]. Arterial embolisation is performed for severe emergency bleeding. Surgical procedures have shown the highest efficiency as they treat the bleeding in its origin; however, they are not exempt from complications such as crusting, bad smell, scarring, and obligatory mouth breathing. The success of these techniques is not complete nor permanent, and surgery is not appropriated in cases of mild or moderate epistaxis [1].

Local sclerotherapy consists of submucosal or subperichondrial injections of polidocanol, an agent causing obstruction of the blood flow and clotting and collapse of the lesion. Sclerotherapy is an established treatment for AVM (varicose veins) and bleedings of the GI and of the genitourinary tract. It has been used in angiomas in the head and neck but is not standardised for HHT. A review of this technique was published by Dr. Morais using submucosal injections of polidocanol. He performed nearly 300 injections in 45 patients for 15 years [8]. Nosebleeds improved both in frequency and quantity in 95% of the cases, and no relevant side effects were reported. More recently, sclerotherapy has been used with satisfactory results by Boyer et al. [6]. Thus, sclerotherapy should be considered as an alternative to surgical procedures, with the advantage of its reduced aggressiveness and low risks derived from it.

## 2. Materials and Methods

### 2.1. Patients and Study Design

This work represents a cross-sectional study performed between 2017 and 2019. It includes HHT patients treated with sclerotherapy at the Hospital Universitario Fundación Alcorcón (HUFA). Before this study, in 2017, we conducted another cross-sectional study on 38 HHT patients treated with sclerotherapy and topical propranolol [9]. That first study was continued and updated in 2019, scaling up to a total of 105 patients. All patients were older than 18 years, had a confirmed diagnosis of HHT (clinical and/or genetic), and were followed by the HUFA Otorhinolaryngology Unit.

Patients underwent a general clinical screening: variables such as gender, frequency of epistaxis, previous and current medical treatment for epistaxis, and the length of time they were under sclerotherapy treatment were recorded. Specific otorhinolaryngologist examination (anterior rhinoscopy and endoscopy) was always performed by the same otorhinolaryngologist specialist in HHT. Telangiectases were described by their morphological characteristics [10] according to Zarrabeitia’s classification: absence of telangiectases (0); isolate punctuated telangiectases (I); multiple punctuated telangiectases (II); ramified telangiectases (III); isolate complex vascular malformations (IVa); multiple complex vascular malformations (IVb).

As inclusion criteria, patients should not have received polidocanol sclerotherapy at least in the 4 weeks prior to the study. The study received approval from the Pharmaceutical Committee and the Clinical Research Ethics Committee from HUFA.

Exclusion criteria included patients under 18 years of age, Young’s complete nasal closure, and allergies to polidocanol or its excipients.

### 2.2. Outcomes and Assessments

The primary outcome was the impact on frequency and severity of epistaxis, as measured by the HHT epistaxis severity score (HHT-ESS). HHT-ESS is an objective, standardised and internationally validated reference tool to estimate the severity of epistaxis in HHT patients [11]. HHT-ESS is calculated from six parameters related to epistaxis: frequency, duration, and intensity of nosebleed, presence/absence of anaemia, need for blood transfusion, and need to seek medical care to treat the haemorrhage.

In this study, the baseline as reference was 4 weeks before starting the treatment and at least 4 weeks after the last session of sclerotherapy. The ESS scale ranges from 0 (absence of epistaxis) to 10. The scores indicates that from 0 to 1 is considered inside the normality range, from 1 to 4 nosebleed is classified as mild, from 4 to 7 is moderate, and from 7 to 10 the haemorrhage is severe, not only searching medical assistance but also being associated with anaemia and requiring blood transfusions. No serious adverse events to the treatment were reported. Any new symptom not present at the baseline was tracked.

Patient’s QOL before and after treatment was assessed by EuroQol-5D-3L (EQ-5D) scale [12]. This scale ranges from 0 (the worst situation—death) to 1 (the best health status). In addition, a visual analogue scale (VAS) which estimates (QOL) with a value from 0 (the worst health status) to 10 (the best health status), was included. For this study, approval from local Clinical Research Ethics Committee was received, and for the inclusion of each patient, informed consent was required.

### 2.3. Sclerotherapy with Polidocanol 1%

Patients were attended at the office, on-demand. The criteria to treat them were recurrent and/or gushing or pouring nosebleeds. Patients contacted the ENT specialist when their nosebleed required care, either for its frequency or its severity. Patients were immediately attended as soon as possible, or with a maximum waiting time of 2 days.

The procedure started with the administration of topical anaesthesia (lidocaine 1% and adrenaline) to numb the area. To minimise the discomfort of the treatment, the anaesthetic is left in place for at least 10 min, usually more. The adrenaline helps not only to decrease the bleeding but it also helps with the identification of the telangiectases: the mucosa becomes paler while the telangiectases stay red due to the impairment of the elastic fibres and the muscular layer on their walls. Once identified, the polidocanol (1%) is injected in small volumes with a 1 mL syringe and a 30 G needle (25 G when the lesions are farther into the nasal cavity and a longer needle is required). The quantity of liquid varies according to the tolerance of the patient and the size of the lesions; usually, the first injections are the most painful, as topical anaesthesia just numbs the area. The procedure (infiltration) is repeated as many times as needed, though never on both sides of the septum at the same time to avoid septum perforations. The administration takes place with different 1 mL syringes and 30 G needles. After the procedure, residual bleeding is controlled with a self-absorbent packing (e.g., Surgicel or similar). Finally, patients receive prophylactic antibiotic treatment, either topically (terramycin ointment, tobramycin or gentamycin drops) or systemically (doxycycline), administered every 12 h for 1 week. Following sclerotherapy, propranolol is used as a topical nasal ointment (0.5% propranolol in Vaseline) prepared at the Hospital Pharmacy as a maintenance treatment to space the time between sclerotherapies.

### 2.4. Statistical Analysis

For data analysis, the SPSS17 program was used. The data are described by absolute and relative frequencies for qualitative variables and by the mean and standard deviation (SD) for quantitative variables according to data distribution. To analyse the change in the dichotomous variables, McNemar asymmetry test was used, and to analyse the change of quantitative variables, the t-Student test of repeated measures, the Chi-square test, and the non-parametric Wilcoxon test were used. All tests are considered bilateral and as significant when the *p*-value < 0.05 or lower.

## 3. Results

A total of 150 HHT patients were treated and evaluated between March 2017 and May 2019, but only 105 completed the study. Missing data were due to a lack of filling the patient’s diary, lack of adherence to treatment, or non-acceptance to enter the study. A total of 59% (*n* = 62) were women, and the average age was 54.6 ± 14.0 years. All HHT patients in the study presented nasal telangiectases and epistaxis (100.0%). At baseline, patients showed either isolated or multiple telangiectases in nostrils. Complex malformations were common in most of our patients. More than 50% of the patients presented grade IVb lesions according to Zarrabeitia’s classification on the nasal mucosa [10].

Due to the extension and the complex macroscopic characteristics of the telangiectases, many patients needed several sclerotherapy sessions to control the symptoms. The duration of each session would depend on the clinic, the complexity and extension of the lesions, the bleeding during the procedure, the endurance to pain and the patient physical and psychological status. The nose is a sensitive organ, and some of the injections, especially those in the lateral wall, the turbinates, or the valve, in spite of topical anaesthesia, are quite painful. The frequency of the visits is also dependent on the bilateral affectation since it is not advised to treat both sides of the septum at the same time.

### 3.1. HHT Epistaxis Severity Score

Baseline mean HHT-ESS pre-treatment was 6.23 ± 2.32 and decreased significantly after-treatment to 1.64 ± 1.6 (*p* < 0.05) (Figure 1A). In the 22 patients with mild ESS, the improvement was 2.1 points (from 2.79 before treatment to 0.7 after sclerotherapy); in the 35 patients with moderate EES, the improvement was 4.12 points (from 5.52 pre to 1.4 post), and in the 47 patients with severe epistaxis, the mean improvement of their ESS achieved 6.1 points (from 8.4 pre- to 2.3 post-treatment). These differences were statistically significant (*p* < 0.001) in all the groups. When all the severity groups were considered together, differences between before and after treatment were also highly significant (Figure 1B).

Then, each of the parameters conforming to the HHT-ESS was individually analysed. All the six factors of the HHT-ESS pre- and post-treatment were significantly reduced.

Regarding epistaxis frequency (Figure 2A), at baseline, 41.9% (44) of patients had several epistaxes daily, 10.5% at least once a day, 31.4% several times per week, 12.4% once a week, and only 4 patients (3.8%) referred to bleedings as once a month. There was no patient referring less than once a month bleeding. However, after the sclerotherapy, only 6 patients (5.7%) still had nosebleeds several times per day, 12 patients (11.4%) several times per week, 25.7% once a week, 24.8% once a month, and 34 patients (32.4%) less than once a month (Figure 2A). The length of the nosebleeds is shown in Figure 2B. Before treatment, bleeding was longer than 30 min in 19% of our patients, 23.8% were bleeding between 16 and 30 min, 28% from 6 to 15 min, (27.6%) from 1 to 5 min, and just 1 patient was bleeding less than a minute. After sclerotherapy, just 1 patient had nosebleeds longer than 15 min, 11 patients (10.5%) from 6 to 15 min, 41 patients (39%) from 1 to 5 min, and 52 patients (49.5%) had less than 1-min nosebleeds (Figure 2B).

The intensity of the nosebleed was described as gushing or pouring by 72 patients (68.6%) and non-gushing by 31.4% before treatment (Figure 3). After treatment, only 6 patients (5.7%) still had gushing nosebleeds, while the majority, 99 patients (94.3%), described their haemorrhages as non-gushing. This difference was statistically significant (*p* < 0.001) (Figure 3). In 67 patients (63.8%), medical attention was needed because of nosebleeds before the sclerotherapy treatment (Figure 4). After the sclerotherapy, only 13 patients (12.4%) needed medical attention (Figure 4). This difference was statistically significant comparing before and after treatment (*p* < 0.001).

Finally, the results concerning anaemia and blood transfusions are shown in Figure 5A. A total of 74 patients (70.5%) were anaemic before treatment (2 in the mild ESS group, 29 in the moderate one, and 43 in the severe group). After the treatment, only 19 patients (18.1%) (4 from the moderate HHT-ESS group and 15 from the severe one) were still anaemic. This difference was statistically significant (*p* < 0.001) (Figure 5A). A number of 31 patients (29.5%), all of them in the severe HHT-ESS group, had received blood transfusions because of anaemia by nosebleeds. After the sclerotherapy, only eight patients (7.6%) still needed blood transfusions (Figure 5B).

### 3.2. Results of the QOL Measurements

QOL was measured by two scales—EQ-5D and VAS. In both cases, the results improved, as shown in Figure 6A,B, respectively. The results measured by the EQ-5D scale improved significantly from 0.7 ± 0.26 before treatment to 0.92 ± 0.16 after treatment *p* < 0.05. Every dimension of the scale increased its value, observing the highest difference in daily activity and psychological parameters such as and anxiety/depression (Figure 6A). When completing the EQ-5D pre-treatment, the patient had to evaluate his/her health status prior to the beginning of the sclerotherapy. For the EQ-5D post-treatment, the patients’ answers had to reflect their current status. The statistical analysis applied was a paired t-Student test between pre- and post-treatment data. The mean difference on the EQ-5D before and after treatment was 0.22 ± 0.25. This difference was statistically significant (*p* < 0.05) with a confidence interval of 95% (IC 0.27 ± 0.17). Taking into account each parameter in the EQ-5D, the following results are observed:

EQ-5D Mobility: at baseline, 25% of patients complained of mobility problems. After the treatment, only 8% still had mobility problems. This difference was statistically significant (chi-square test *p* < 0.0001).

EQ-5D personal care: at baseline, 16% of patients had some trouble dressing or washing. After treatment, only 3% referred problems. This difference was statistically significant (chi-square test *p* < 0.0001).

EQ-5D daily activities: before treatment, 9% of patients declared they were unable to perform normal daily activities, and 51% had some kind of impairment. After the sclerotherapy, only 1% was still unable to complete their chores, and 13% had some impairment, while 86% of the patient referred no trouble at all. This difference was statistically significant (chi-square test *p* < 0.0001).

EQ-5D pain/discomfort: at baseline, 4% of patients complained of severe pain or discomfort and 40% of patients referred moderate pain or discomfort. After treatment, just 1% still had severe pain/discomfort and 12% moderate pain/discomfort; 87% of patients had no pain or discomfort. These answers showed an improvement of 67%, which was statistically significant (chi-square test *p* < 0.001).

EQ-5D anxiety/depression: at baseline, 21% of patients were very anxious or depressed, and 34% referred to moderate anxiety or depression. After treatment, 2% still had severe anxiety/depression and 17% still were moderately anxious/depressed; 81% of patients were fine. A total of 41% of the severe cases improved to moderate, and 50% of the moderate anxious/depressed patients had a complete recovery. This difference was statistically significant (chi-square test *p* < 0.001).

VAS is a descriptive self-evaluation scale of health status. VAS is also designed to measure QOL, 0 being the worst and 10 being the best score. An increase in the VAS scale from 4.48 ± 2.4 before to 8.35 ± 1.25) after the treatment was observed, and it was statistically significant (*p* < 0.05) (Figure 6B). The mean difference on the VAS scale before and after treatment was 3.87 ± 2.36, 1.81 ± 1.5 in the mild group, 3.5 ± 2.02 in the moderate group, and 4.12 ± 1.48. This difference was statistically significant (*p* < 0.001) for all the groups.

## 4. Discussion

The results presented in this study s demonstrate that on-demand sclerotherapy with polidocanol is a highly effective therapy, not only in reducing the severity of bleedings in the HHT population according to the HHT-ESS score but also in improving their QOL. All patients showed a drop in the HHT-ESS value before and after the sclerotherapy.

The minimal important difference (MID) of ESS in HHT patients was calculated by Yin et al. [13] as 0.71 to be clinically significant. In our study, the mean decrease achieved was 4.58 ± 2.3 points (2.1 in our 22 mild epistaxis group, 4.1 in the 36 patients with moderate nosebleeds, and 6.1 in the 47 subjects with severe bleedings). Interestingly, this decrease represents the highest ESS score drop recorded until now in the literature in a large group of patients.

Prior to treatment, most of our patients were classified as suffering from severe epistaxis according to their HHT-ESS. More than 90% of them (*n* = 101) had more than one bleeding episode per week. Among them, 41.9%, referred several episodes per day. These data are similar to the baseline score in the Whitehead et al. [14] cohort, who participated from 2011 to 2015 in the NOSE clinical trial study, to evaluate the safety and efficacy of different nasal sprays (bevacizumab 1%, estriol 0.1%, tranexamic 10%, and saline serum as placebo). In Whitehead et al., no significant differences were observed before and after treatments compared to placebo, according to the HHT-ESS. However, in our work, all the variables composing the HHT-ESS scale were significantly decreased. After the sclerotherapy procedure, only six patients continued bleeding several times a day.

The EQ-5D is a valid method to assess the impact of nosebleeds on the QOL of HHT patients. Our results show that those items with a larger number of affected patients before treatment (mainly daily activities and anxiety/depression) were the ones improving more after treatment. The same holds true for the VAS scale. A linear correlation between the HHT-ESS decrease and the increase in the QOL scales is detected.

The recovery of the QOL in our patients, evaluated by both EQ-5D and VAS, is remarkable. The anxiety/depression item is the dimension showing a better improvement in our patients after the treatment. The majority (58%) of patients referred psychological problems at baseline, and only 18 (19%) complained about their emotional state after the treatment. This improvement is in line with the reduction of the HHT-ESS. Bleeding and weakness due to anaemia affect the anxiety/depression state. The decrease in bleeding and the improvement in the anaemic condition helps to strengthen the psychological state of the patients.

The baseline QOL score in the Zarrabeitia et al. [15] cohort was better than in our population. The dimensions of daily activities and anxiety/depression of EQ-5D were impaired in 60% and 54.3%, respectively, in our patients was 24.6%, and 43.9% in the Zarrabeitia et al. [15] group. Moreover, baseline VAS values scored 7.37 points, while our baseline VAS mean value was 4.48 points. Thus, even when the QOL of our cohort was more severely affected, the improvement following sclerotherapy was remarkable.

Boyer et al. [6] performed sclerotherapy with sodium tetradecyl sulphate in a cross-sectional study evaluated for a period of 12 weeks. A significant reduction of 0.95 points on the HHT-ESS scale (*p* = 0.027) was observed. The fact that our results are far better may be attributed to the time of follow-up. Patients may require several sessions of sclerotherapy to achieve good control of their haemorrhages, and 12 weeks might not be enough. A longer follow-up allows the ENT to monitor the appearance of new lesions and to perform new sclerotherapy procedures on-demand upon nosebleeds recurrences.

Side effects of sclerotherapy in our group are either negligible or non-existing. If we compare our study to surgical procedures, the risk/benefit complications are highly favourable in our case. With drugs, such as thalidomide, the safety profile is rather unfavourable for thalidomide. Thalidomide cannot be administered without interruption due to its adverse events, peripheral neuropathy, among others [16]. On the other hand, upon treatment discontinuation, a considerable rebound effect appears.

In the case of bevacizumab, the cost-effectiveness is favourable to polidocanol, though the injection of bevacizumab and cyanoacrylate glue sounds promising [17], showing improvement in 31 patients with moderate-severe epistaxis after a mean follow up of 26.6 months (interval from 9 to 56). From a baseline HHT-ESS of 7.8 before injection, it decreased to 3.8 after the treatments. It is a line of treatment worth watching.

After more than 8 years of sclerotherapy practice in HHT patients, we must say that in our hands, the treatment is safe, with good tolerance and with hardly any relevant adverse effects.

### Limitations of the Study

This study has important limitations. Clinical manifestations are distinct in different HHT patients, and they usually worsen with age. Epistaxis is the most common symptom, but there are many more (pulmonary involvement, digestive issues, hepatic lesions, strokes, and internal bleeding). We have only evaluated the changes related to the ENT field. Patients demanded the otorhinolaryngology consultation, and their treatment was scheduled on-demand based on their epistaxis symptoms. The distance to the hospital represents one limitation, especially in severe cases or older patients, when travelling becomes a challenge.

## 5. Conclusions

Sclerotherapy does not represent a definite cure for nosebleeds. However, it supposes an improvement not only on epistaxis but also on clinically related issues such as anaemia and psychological status) and, consequently, on the QOL.

Many patients manifest their satisfaction overtly on the treatment and the positive influence on their lives. This study demonstrates that on-demand sclerotherapy performed at the office significantly reduces HHT epistaxis and greatly improves the patients’ QOL.

## Figures and Tables

**Figure 1 jcm-10-03845-f001:**
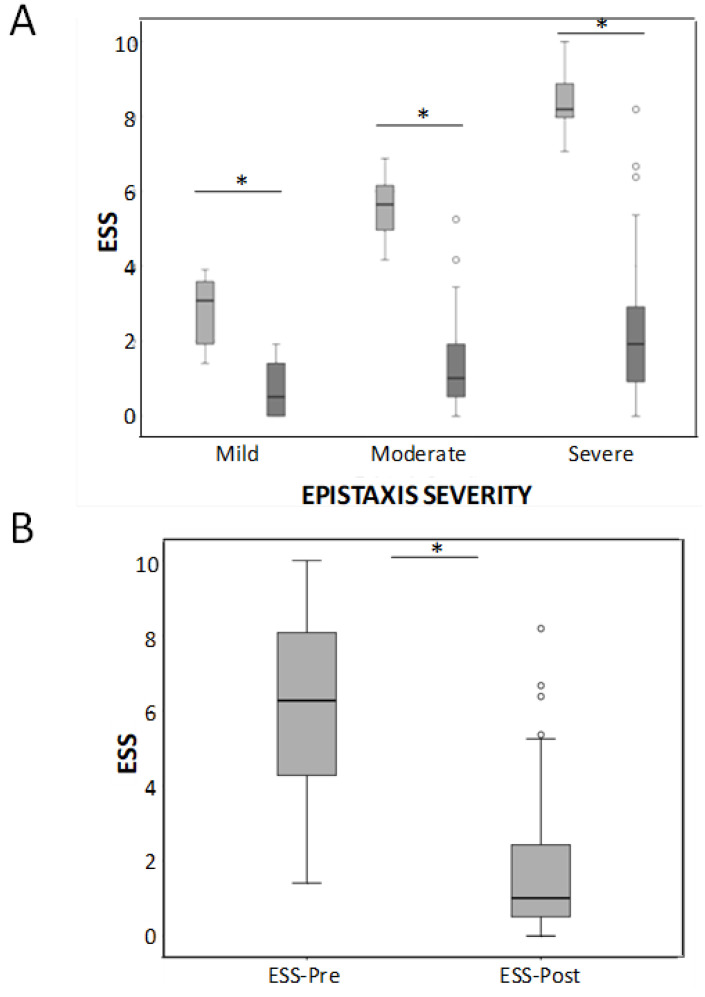
(**A**) Box-whiskers plots corresponding to the HHT-ESS in mild, moderate, and severe HHT patients before and after the sclerotherapy treatment. * *p* < 0.001. (**B**) Considering all the severity groups together, differences between before and after treatment were also highly significant * *p* < 0.001.

**Figure 2 jcm-10-03845-f002:**
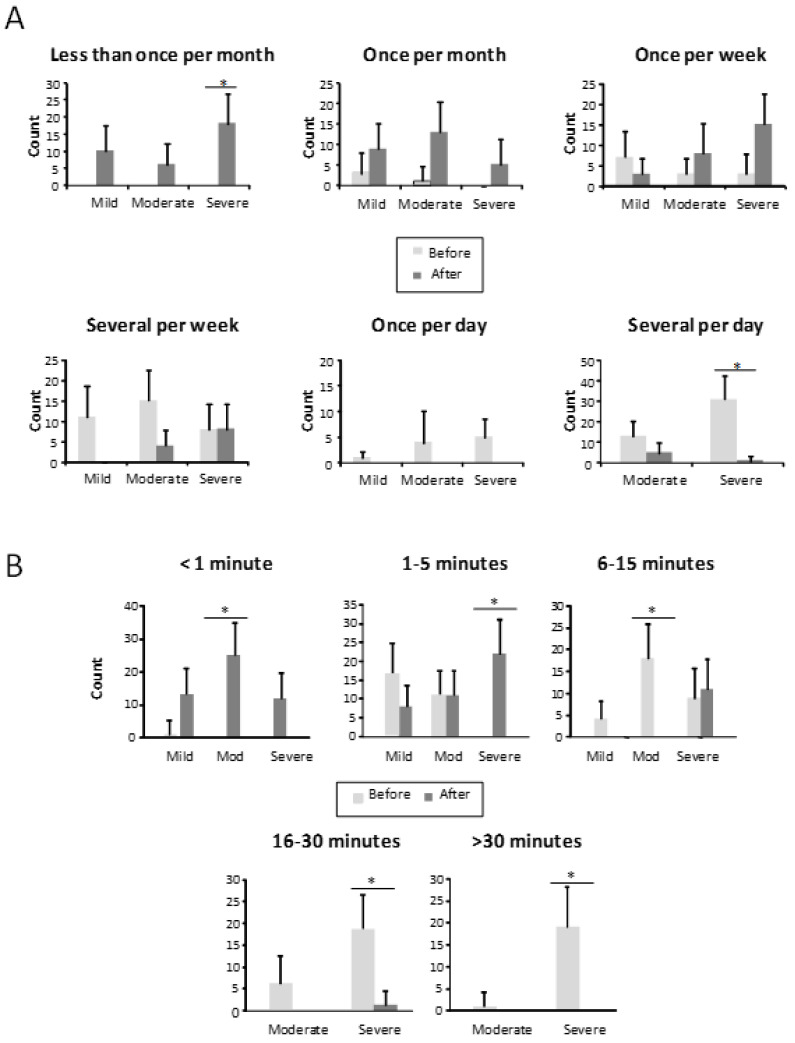
(**A**) Frequency of epistaxis, in the different groups, before and after the treatment in mild, moderate, and severe HHT-ESS patients. * *p* < 0.05. (**B**) Bleeding time of patients in each class, before and after the treatment in mild, moderate, and severe HHT-ESS patients. * *p* < 0.05.

**Figure 3 jcm-10-03845-f003:**
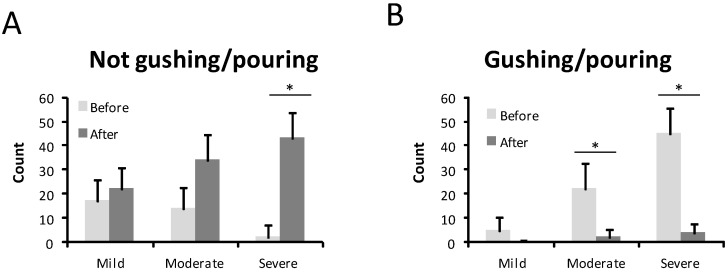
Type of bleeding before and after the treatment in mild, moderate, and severe HHT-ESS patients. (**A**). Not gushing /pouring. (**B**). Gushing /pouring. * *p* < 0.05.

**Figure 4 jcm-10-03845-f004:**
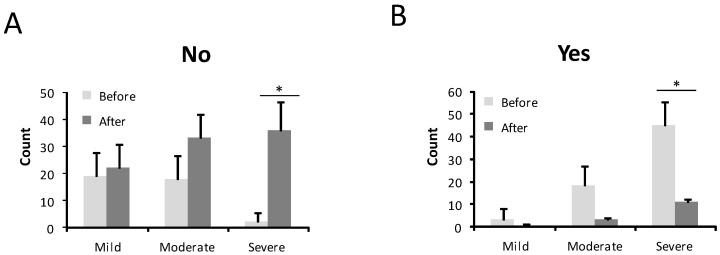
Number of patients who needed medical care before and after the treatment in mild, moderate, and severe HHT-ESS patients. (**A**). No need medical care. (**B**). Need medical care. * *p* < 0.05.

**Figure 5 jcm-10-03845-f005:**
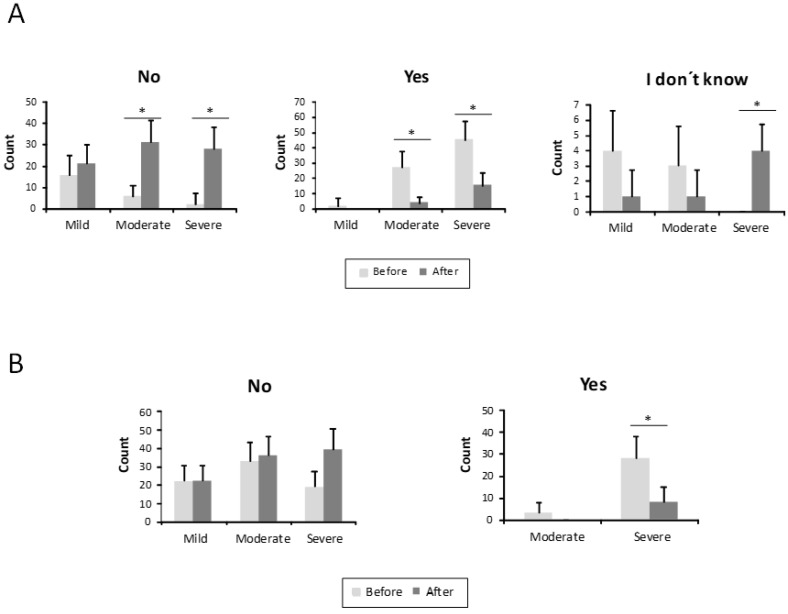
Parameters related to anaemia from the HHT-ESS. (**A**) Number of patients is represented with, without anaemia, or without anaemia test, before and after the treatment in mild, moderate, and severe HHT patients. (**B**) The need for transfusions in the same groups of patients is shown before and after the treatment. * *p* < 0.05.

**Figure 6 jcm-10-03845-f006:**
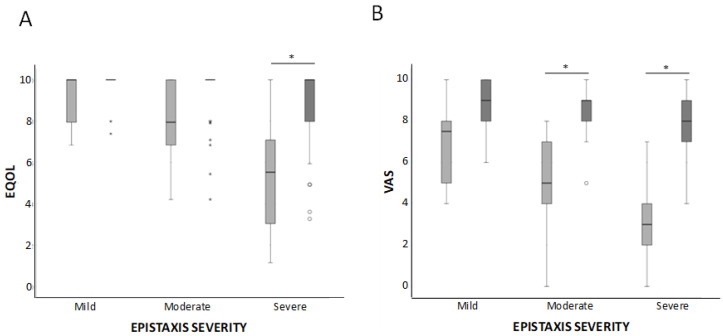
Box-whiskers plots corresponding to QOL parameters, EQ-5D (**A**) and VAS (**B**). Results are shown before and after the treatment in mild, moderate, and severe HHT-ESS patients. * *p* < 0.05.

## Data Availability

Reported results can be found in the files of Centro de Investigaciones Biológicas Margarita Salas (CIB, CSIC) and in the files of Hospital Universitario Fundación Alcorcón (HUFA), Madrid, Spain.

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
