# Peer review of "Sclerotherapy on Demand with Polidocanol to Treat HHT Nosebleeds"

_jcm, 2021, doi:10.3390/jcm10173845_

Round 1

Reviewer 1 Report

 In the introduction the authors state that :

"Recurrent or severe epistaxis should be referred to an otorhinolaryngologist..."  and 

"Treatments are applied depending to the frequency and severity of epistaxis, as determined by the clinician´s experience. "

it seems to me that these statements are conflicting with the "on demand" approach. The introduction can be shortened to avoid some repetitions.

For mildly severe recurrent.... line 78, not clear, mild or severe? 

After nearly 300 injections.... line 95, not clear, how many/patient? what is the time interval??, 

 polidocanol    check spelling, polydocanol or polidocanol??

 The patients are attended at the office, on demand, in base on their symptoms.    line 147, please make it more clear: the authors mean that the  patients when evaluated at the center did not receive any indication as to the best therapy for their nasal condition? was only patient's decision the sclerosing therapy??  

for at least 10 to 15 minutes, usually more....line 150, define the effective time that was used

the tolerance of the patient and the size of the lesions, usually the first injections are the most painful...  line 156, the anaesthesia does not work? please clarify or discuss

never on both sides of the septum at the same time to avoid septum perforations...line 158 and following statements about antibiotics; the statements look certainly correct, but they stess the presence of possible relevant complication of the procedure, the authors should clarify why the procedure is "non demand" and not after recognising a medical indication 

lines 190to 195: the data reported in the text do not fit  what is in fig 1; before and after is not indicated

figures 2, 3, 4, 5 the details to refer to the txt (ABC and so on, are not included)

in general the authors should avoid repeating in the text data provided in figures.

the results presentation is not immediately straifghtforward,  in some figures condition after treatment seems worse than before after a glance and the reader needs to read several times to catch the message. 

minor:  several misstyping 

Author Response

Reviewer 1.

  1. In the introduction the authors state that :

"Recurrent or severe epistaxis should be referred to an otorhinolaryngologist..."  and 

"Treatments are applied depending to the frequency and severity of epistaxis, as determined by the clinician´s experience. "

it seems to me that these statements are conflicting with the "on demand" approach. The introduction can be shortened to avoid some repetitions.

Answer

Thank you very much for your observation. Introduction has been shortened to avoid some repetitions as you state, changes has been highlighted to make an easier follow-up. Furthermore, the statements mentioned by the reviewer, have been fused in one sentence, as a general recommendation in the introduction,

Page 2 at the end of the third paragraph:

“Recurrent or severe epistaxis should be referred to an otorhinolaryngologist for local control, and followed by blood tests to evaluate the need of oral or IV iron supplementation or blood transfusions to treat anaemia”.

The “on demand” approach with sclerotherapy is described and discussed in the following sections: materials and methods, results and discussion.

  1. For mildly severe recurrent.... line 78, not clear, mild or severe?

Answer: We have changed the statement as: “recurrent or severe epistaxis”

  1. After nearly 300 injections.... line 95, not clear, how many/patient? what is the time interval??,

Answer: We have clarified the point with more specific details:

“He performed  nearly 300 injections in 45 patients during 15 years.”

  1. polidocanol    check spelling, polydocanol or polidocanol??

Answer: Thank you very much for your comment. Consulting the references about the term in PubMed, we find the word written as both, polidocanol and polydocanol. However, for the sake of uniformity, we have adopted the term polidocanol, all over the manuscript

  1. The patients are attended at the office, on demand, in base on their symptoms.    line 147, please make it more clear: the authors mean that the  patients when evaluated at the center did not receive any indication as to the best therapy for their nasal condition? was only patient's decision the sclerosing therapy??

Answer:

Authors are grateful to the reviewer for the help in clarifying the issue. We have written the following sentence, in page 4 section 2.3

The patients were attended at the office, on demand. Patients contact the ENT specialist when their nosebleed requires care either for its frequency or its severity. They are seen as soon as possible. Sometimes they are attended on the same day, though an appointment within a couple of days is preferred.

  1. for at least 10 to 15 minutes, usually more....line 150, define the effective time that was used

Answer:

We have defined the time as “for at least 10 minutes”, page 5 at the top second paragraph.

  1. The tolerance of the patient and the size of the lesions, usually the first injections are the most painful...  line 156, the anaesthesia does not work? please clarify or discuss

Answer: Thank you, we have clarified this issue in red, page 5.

“usually the first injections are the most painful, “as topical anaesthesia just numbs the area.”

8.never on both sides of the septum at the same time to avoid septum perforations...line 158 and following statements about antibiotics; the statements look certainly correct, but they stess the presence of possible relevant complication of the procedure, the authors should clarify why the procedure is "non demand" and not after recognising a medical indication 

Thank you for your observation. We have clarified the medical indication adding:

Section 2.3 page 4

“The criteria to treat them are: recurrent and/or gushing or pouring nosebleeds.”

9.lines 190to 195: the data reported in the text do not fit  what is in fig 1; before and after is not indicated figures 2, 3, 4, 5 the details to refer to the txt (ABC and so on, are not included)

in general the authors should avoid repeating in the text data provided in figures.

the results presentation is not immediately straightforward,  in some figures condition after treatment seems worse than before after a glance and the reader needs to read several times to catch the message. 

Answer:

Thank you very much for your comments and impressions. We have tried to improve this issue by:

  • adding a part B to figure 1, when the total results, before, vs after, are compared by joining the different types of bleeding severity. We have added A and B.
  • the former figures 2 and 3 are now Figure 2, since it describes the mainly characteristics of the bleeding: frequency (A) and time of bleeding (B) .
  • We have deleted the letters of the text, since the different categories of each item are self-explained in the figures on top of each graphic.

 We really hope this time the reader may follow the results in an easier way.

  1. minor:  several misstyping 

Answer:

Thank you. We have tried to correct the misstyping and other English mistakes.

Reviewer 2 Report

The authors describe the effect of sclerotherapy. The results are interesting and important for HHT patients en physicians treating HHT patients.

I have several remarks / questions.

  1. The English should be improved. There are far too many language errors. I stopped making corrections at line 70....
  2. The legend below the figures mentions HHT severity. That should be Epistaxis severity. An HHT severity scale has been developed, but the authors describe epistaxis severity, NOT HHT severity. 
  3. Figures 2 and 3 are not clarifying, but confounding. They should be combined, figuring mean duration  of epistaxis before and after treatment, or something like that.
  4. In the abstract it is mentioned that the sclerotherapy is combined with topical propranolol, but topical propranolol is not mentioned at al in the method section and discussion. Please explain.

Author Response

Reviewer 2

Comments and Suggestions for Authors

The authors describe the effect of sclerotherapy. The results are interesting and important for HHT patients en physicians treating HHT patients.

I have several remarks / questions.

  1. The English should be improved. There are far too many language errors. I stopped making corrections at line 70....

Answer: Thank you very much. We have revised the text with edition of all the typos and English style.

  1. The legend below the figures mentions HHT severity. That should be Epistaxis severity. An HHT severity scale has been developed, but the authors describe epistaxis severity, NOT HHT severity. 

Answer: This is a very interesting point, that we have taken into account. Now, we have added HHT-ESS since the reviewer is completely right. Thank you very much.

  1. Figures 2 and 3 are not clarifying, but confounding. They should be combined, figuring mean duration of epistaxis before and after treatment, or something like that.

Answer: Thank you for the suggestion. We have combined both figures, as they represent the two main traits of the epistaxis: frequency and duration of the bleeding. Now this is Figure 2, divided in part A and Part B, A represents frequency, and B time of bleeding. On top of each graphic the corresponding category is specified

  1. In the abstract it is mentioned that the sclerotherapy is combined with topical propranolol, but topical propranolol is not mentioned at al in the method section and discussion. Please explain.

Answer: Thank you  for your comment. We have added the treatment, at the end of the method section 2.3, page 5:

Following sclerotherapy, propranolol is used as a topical nasal ointment (0.5% propranolol in vaseline) prepared at the Hospital Pharmacy, as a maintenance treatment to space the time between sclerotherapies.

This manuscript is a resubmission of an earlier submission. The following is a list of the peer review reports and author responses from that submission.